# PREDICTIVE LOCAL SMOOTHNESS FOR STOCHASTIC GRADIENT METHODS

## ABSTRACT

Stochastic gradient methods are dominant in nonconvex optimization especially for deep models but have low asymptotical convergence due to the fixed smoothness. To address this problem, we propose a simple yet effective method for improving stochastic gradient methods named predictive local smoothness (PLS). First, we create a convergence condition to build a learning rate which varies adaptively with local smoothness. Second, the local smoothness can be predicted by the latest gradients. Third, we use the adaptive learning rate to update the stochastic gradients for exploring linear convergence rates. By applying the PLS method, we implement new variants of three popular algorithms: PLS-stochastic gradient descent (PLS-SGD), PLS-accelerated SGD (PLS-AccSGD), and PLS-AMSGrad. Moreover, we provide much simpler proofs to ensure their linear convergence. Empirical results show that the variants have better performance gains than the popular algorithms, such as, faster convergence and alleviating explosion and vanish of gradients.

## 1 INTRODUCTION

In this paper, we consider the following nonconvex optimization:

$$\min_{x \in \mathbb{R}^d} f(x) := \frac{1}{n} \sum_{i=1}^{n} f_i(x), \tag{1}$$

where $x$ is the model parameter, and neither $f$ nor the individual $f_i$ ($i \in [n]$) are convex, such as, deep models. Stochastic gradient descent (SGD) is one of the most popular algorithms for minimizing the loss function in equation 1. It iteratively updates the parameter by using the product of a learning rate and the negative gradient of the loss, which is computed on a minibatch drawn randomly from training set. Unfortunately, small learning rate makes SGD painfully slow to converge, and high learning rate causes SGD to diverge. Therefore, choosing a proper learning rate (or step size) becomes a challenge.

A popular adaptive method adjusts automatically the learning rate by using some forms of the past gradients to scale coordinates of the gradient. For example, AdaGrad Duchi et al. (2011) is the first adaptive algorithm to update the sparse gradients by dividing positive square root of averaging the squared past gradients, and its several variants (e.g., Adadelta Zeiler (2012), RMSProp Tieleman & Hinton (2012), Adam Kingma & Ba (2015), Nadam Dozat (2016), and AMSGrad Reddi & Kumar (2018)) have been widely and successfully applied into train deep models. Specifically, they use the exponential moving averages of squared past gradients to manage the rapidly decayed learning rate. Most of these algorithms are to establish their convergence guarantees based on a maximum of the Lipschitz constant $L$, which governs smoothness of the loss function in whole parameter space, often called *L-smoothness* Bottou et al. (2016); Reddi et al. (2016).

However, the maximum of the $L$-smoothness results in a small learning rate as it is inversely proportional to $L$[1] Bottou et al. (2016). Since the $L$-smoothness changes with the parameter and the given data, it leads to a local smoothness of the loss function near the optimum for selecting the learning rate. Specifically, the local smoothness is often computed by the the Hessian matrix $\nabla^2 f(x)$, such as data dependent local smoothness (e.g., local SVRG Vainsencher et al. (2015)) and parameter dependent local smoothness (e.g., Newton method Nocedal & Wright. (2006)). Unfortunately, the Hessian matrix results in a high computational cost and a high memory requirement. Recently,

---

[1]A large $L$ results in a low learning rate to slowly move the loss function from a point $f(x_0)$ to local minimum loss $f(x^*)$ with an equilibrium parameter $x^*$, and vice versa.

Stochastic Quasi-Newton (SQN) Byrd et al. (2016) and Adaptive Quasi-Newton (adaQN) Keskar & Berahas (2016a) are to partly reduce the expensive cost by using the Hessian-vector product Nocedal & Wright. (2006). However, these algorithms are still difficult to train a large number of the parameters of deep models due to the loop computation of the Hessian-vector product.

To address the above problem, we exploit a simple yet effective method to predict the parameter dependent local smoothness for the adaptive learning rate. To ensure the convergence of the stochastic algorithms, an ideal local smoothness $\mathcal{L}(x_t)$ of the loss function is based on an ideal neighborhood $N_{r_{x_t}}(x^*)$ of an equilibrium $x^*$ with a radius $\|x_t - x^*\|$ by using the current parameter $x_t$. However, it is difficult to apply the $N_{r_{x_t}}(x^*)$ into the stochastic algorithms due to the unknown equilibrium $x^*$. Since the past parameter $x_{t-1}$ has been known, we consider a local smoothness $L(x_t)$ on a neighborhood $N_{r_{x_{t-1}}}(x_t)$ of $x_t$ with a radius $\|x_t - x_{t-1}\|$. Because $N_{r_{x_t}}(x^*)$ and $N_{r_{x_{t-1}}}(x_t)$ have a common point $x_t$ or region, $L(x_t)$ is naturally used instead of $\mathcal{L}(x_t)$. Based on the definition of the $L$-smoothness (or Lipschitz continuous gradients) Bottou et al. (2016), $L(x_t)$ on this neighborhood is easily predicted by using the past gradient $\nabla f(x_{t-1})$ and the current gradient $\nabla f(x_t)$, that is, $L(x_t) = \frac{\|\nabla f(x_t) - \nabla f(x_{t-1})\|_2}{\|x_t - x_{t-1}\|_2}$. Obviously, it only needs to storage the past parameter $x_{t-1}$ and gradient $\nabla f(x_{t-1})$, and perform twice subtractions and norm computations.

This paper, therefore, provides a local smoothness strategy to study the adaptive learning rate. In this strategy there are two important problems that how to easily build a direct functional relationship between the learning rate and the local smoothness, and how to calculate the local smoothness. To address these problems, the stochastic gradient algorithms are transformed into a linear dynamical system Lessard et al. (2016); Hu et al. (2017) by using the local smoothness to linearize the gradient function. The functional relationship is obtained by constructing the convergence condition for the linear dynamical system. Based on the above discussion, the local smoothness is simply predicted by using the current gradient and the latest gradient. Overall, our main contributions are summarized as

- We propose a predictive local smoothness (PLS) method to adjust automatically learning rate for stochastic gradient algorithms. Our PLS method will lead these algorithms to drive a loss function to fast converge to a local minimum.

- We apply PLS into SGD, SQN Byrd et al. (2016), AMSGrad Reddi & Kumar (2018) and AccSGD Kidambi et al. (2018). Correspondingly, we establish four PLS-SGD, PSL-SQN, PLS-AMSGrad and PLS-AccSGD algorithms, and provide corresponding theoretical conditions to ensure their linear convergence.

- We also provide an important empirical result that PLS can alleviate the exploding and vanishing gradients in the classical algorithms (e.g., SGD, AMSGrad Reddi & Kumar (2018) and AccSGD Kidambi et al. (2018)) for training deep model with least squares regression.

Note that due to the limited space PLS-AccSGD is provided in subsection 6.1 in Supplementary Material. We do not provide PSL-SQN since it is similar to PLS-SGD. But, we empirically verify that PSL-SQN is faster than SQN in the subsection 4.1.

## 2 PRELIMINARIES

In this section, we introduce some notations, local smoothness assumption, and three popular stochastic gradient algorithms: SGD, SQN Byrd et al. (2016), AMSGrad Reddi & Kumar (2018), and AccSGD Kidambi et al. (2018).

**Notation.** $\nabla f(x)$ denotes exact gradient of $f$ at $x$, while $\nabla f_{i_t}(x)$ denotes a stochastic gradient of $f$, where $i_t$ is sampled uniformly at random from $[1, \cdots, n]$ and $n$ is the number of samples. Since $i_t$ is sampled in an independent and identically distributed (IID) manner from $\{1, \cdots, n\}$, the expectation of smoothness $L_{i_t}(x_t)$ is denoted as $\mathbb{L}(x_t) = \mathbb{E}[L_{i_t}(x_t)] = \frac{1}{n}\sum_{i=1}^{n} L_i(x_t)$. A $\ell_1$ or $\ell_2$-norm of a vector $x$ is denoted as $\|x\|$, and its square is $\|x\|^2$. A positive-definite matrix $A$ is denoted as $A \succ 0$. The Kronecker product of matrices $A$ and $B$ is denoted as $A \otimes B$. We denote a $d \times d$ identity matrix by $I_d$. A neighborhood of a point $x_1 \in \mathbb{R}^d$ with radius $r_{x_2}$ is denoted as $N_{r_{x_2}}(x_1) = \{y \in \mathbb{R}^d | \|x_1 - y\| < r_{x_2} = \|x_1 - x_2\|\}$.

**Assumption 1.** *We say $f$ is local smoothness on a set $C \subset \mathbb{R}^d$ if there is a constant $L$ such that*

$$\|\nabla f(x) - \nabla f(y)\| \leq L\|x - y\|, \quad for \ \forall \ x, y \in C. \tag{2}$$

Assumption 1 is an essential foundation for convergence guarantees of most stochastic gradient methods as the gradient of $f$ is controlled by $L$ with respect to the parameter vector.

**SGD.** Stochastic Gradient Descent (SGD) simply computes the gradient of the parameters by uniformly randomly choosing a single or a few training examples. Its update is given by

$$x_{t+1} = x_t - \eta_t \nabla f_{i_t}(x_t), \tag{3}$$

where $\eta_t$ is the learning rate. $\eta_t$ is usually set to a decay form $\eta_0/\sqrt{t}$ in practice. This setting leads to slower convergence. Moreover, the gradient of the loss of a deep model with rectified linear units (ReLU) Nair & Hinton (2010) often explodes when large initialization $\eta_0$.

**SQN.** Stochastic Quasi-Newton (SQN) Byrd et al. (2016) employs iterations of the form

$$x_{t+1} = x_t - \eta_t H_t \nabla f_{i_t}(x_t), \tag{4}$$

where $H_{t+1} = (I - \rho_k s_k y_k^T) H_t (I - \rho_k y_k s_k^T) + \rho_k s_k s_k^T$ with $s_k = x_t - x_{t-1}$, $y_k = \nabla f_{i_t}(x_t) - \nabla f_{i_{t-1}}(x_{t-1})$ and $\rho_k = y_k^T s_k$. In practice, the quasi-Newton matrix $H_{t+1}$ is not formed explicitly, and is computed by the Hessian-vector product Nocedal & Wright. (2006).

**AMSGrad.** AMSGrad Reddi & Kumar (2018) is an exponential moving average variant of the popular Adam algorithm Kingma & Ba (2015) in the scale gradient method. AMSGrad uses the factors $\beta_{1t} = \beta_1/t$ and $\beta_2$ to exponentially move the momentums of the gradient and the squared gradient, respectively. $\beta_{1t} = \beta_1 = 0.9$ and $\beta_2 = 0.999$ are typically recommended in practice. The key update is described as follows:

$$\begin{cases} m_{t+1} = \beta_{1t} m_t + (1 - \beta_{1t}) \nabla f_{i_t}(x_t), \\ v_{t+1} = \beta_2 v_t + (1 - \beta_2) \left( \nabla f_{i_t}(x_t) \right)^2, \\ \widehat{v}_{t+1} = \max\{v_{t+1}, \widehat{v}_t\}, \quad x_{t+1} = x_t - \eta_t \frac{m_{t+1}}{\sqrt{\widehat{v}_{t+1}}}. \end{cases} \tag{5}$$

**AccSGD.** Accelerated SGD (AccSGD) proposed in Jain et al. (2017) is much better than SGD, HB Polyak (1964) and NAG Nesterov (1983) in the momentum method. An intuitive version of AccSGD is presented in Kidambi et al. (2018). Particularly, AccSGD takes three parameters: learning rate $\eta_t$, long learning rate parameter $\kappa \geq 1$, statistical advantage parameter $\xi \leq \sqrt{\kappa}$, and $\alpha = 1 - 0.7^2 \xi/\kappa$. This update can alternatively be stated by:

$$\begin{cases} m_{t+1} = \alpha m_t + (1 - \alpha) \left( x_t - \frac{\kappa \eta_t}{0.7} \nabla f_{i_t}(x_t) \right), \\ x_{t+1} = \frac{0.7}{0.7 + (1 - \alpha)} \left( x_t - \eta_t \nabla f_{i_t}(x_t) \right) + \frac{1 - \alpha}{0.7 + (1 - \alpha)} m_{t+1}. \end{cases} \tag{6}$$

## 3 PREDICTIVE LOCAL SMOOTHNESS

The popular adaptive learning rate methods are based on using gradient updates scaled by square roots of exponential moving averages of squared past gradients Reddi & Kumar (2018). These methods indirectly adjust the learning rate as they can be essentially viewed as gradient normalization. In this section, we study the local smoothness to directly and adaptively adjust the learning rate, propose a predictive local smoothness (PLS) method, and apply this method into SGD, AMSGrad Reddi & Kumar (2018), and AccSGD Kidambi et al. (2018). Before showing our PLS method, we first give two local smoothness sequences definition.

**Definition 1.** *Let $x^*$ be an equilibrium of the local minimum $f(x^*)$ and $\{x_t\}_{t \geq 0}$ be a updating parameter procedure, where $x_0$ is an initial point. A corresponding neighborhood sequence of $x^*$ is denoted by $\{N_{r_{x_t}}(x^*)\}_{t \geq 0}$, where $r_{x_t} = \|x^* - x_t\|$. An **ideal local smoothness sequence** of $x^*$ is defined as $\{\mathcal{L}(x_t)\}_{t \geq 0}$ which satisfies that,*

$$\|\nabla f(x^*) - \nabla f(y)\| \leq \mathcal{L}(x_t)\|x^* - y\|, \ for \ \forall \ y \in N_{r_{x_t}}(x^*). \tag{7}$$

*A backward neighborhood sequence on $\{x_t\}_{t \geq 1}$ is denoted by $\{N_{r_{x_{t-1}}}(x_t)\}_{t \geq 1}$, where $r_{x_{t-1}} = \|x_t - x_{t-1}\|$. A **backward local smoothness sequence** is defined as $\{L(x_t)\}_{t \geq 1}$ which satisfies that*

$$\|\nabla f(x_t) - \nabla f(y)\| \leq L(x_t)\|x_t - y\|, \ for \ \forall \ y \in N_{r_{x_{t-1}}}(x_t). \tag{8}$$

This definition reveals that $\mathcal{L}(x_t)$ governs the ideal local smoothness between the equilibrium $x^*$ and $x_t$ to strictly ensure the convergence of the updating parameter procedure. However, $\mathcal{L}(x_t)$ cannot be computed due to the unknown $x^*$. Clearly, $L(x_t)$ is easily calculated by using $x_t$ and $x_{t-1}$.

---

**Procedure PLS**

- Build the learning rate $\eta = \eta(L(x_t))$ by linearizing $\nabla f(x_t)$.
- Predict the local smoothness $L(x_t) = \frac{\|\nabla f(x_t) - \nabla f(x_{t-1})\|}{\|x_t - x_{t-1}\| + \epsilon_1}$.
- Apply $\eta(L(x_t)) \propto \frac{1}{L(x_t) + \epsilon_2}$ to update $x_{t+1}$ by any stochastic gradient algorithms.

---

Figure 1: Predictive Local Smoothness

### 3.1 PLS METHOD

PLS is an adaptive learning rate method based on the local smoothness. The local smoothness varies based on the updating parameters $\{x_t\}_{t \geq 0}$ in stochastic gradient algorithms. According to the definition 1, the local smoothness sequence $\{\mathcal{L}(x_t)\}_{t \geq 0}$ are ideally used to adjust the learning rate in the neighborhood sequence $\{N_{r_{x_t}}(x^*)\}_{t \geq 0}$. However, it is difficult to compute $\{\mathcal{L}(x_t)\}_{t \geq 0}$ because of the unknown $x^*$. Since there is a common point sequence on $\{x_t\}_{t \geq 1}$ in both $\{N_{r_{x_{t-1}}}(x_t)\}_{t \geq 1}$ and $\{N_{r_{x_t}}(x^*)\}_{t \geq 0}$, $\{\mathcal{L}(x_t)\}_{t \geq 1}$ can be predicted by $\{L(x_t)\}_{t \geq 1}$, which is easily computed by using the current gradient and the latest gradient. In this paper, our key idea is to use the predictive local smoothness sequence $\{L(x_t)\}_{t \geq 1}$ instead of the unknown $\{\mathcal{L}(x_t)\}_{t \geq 0}$ to adjust automatically the learning rate $\eta_t$. PLS is described as the following three steps.

**Building adaptive learning rate with local smoothness.** We firstly create a functional relationship between $\eta_t$ and $L(x_t)$ by using the convergence conditions of stochastic gradient algorithms. Following the local smoothness in equation 7, since $\nabla f(x^*) = 0$, $\nabla f(x_t)$ is linearized by $\nabla f(x_t) = (\mathcal{L}(x_t) \otimes I_d)(x_t - x^*)$. Using $L(x_t)$ instead of $\mathcal{L}(x_t)$, we consider the following linearization

$$\nabla f(x_t) = (L(x_t) \otimes I_d)(x_t - x^*), \tag{9}$$

where $L(x_t)$ is computed by equation 10 in the neighborhood $N_{r_{x_{t-1}}}(x_t)$, and $L(x_t) = \|\nabla^2 f(x_t)\|$ if $f$ is twice continuously differentiable. Stochastic gradient algorithms can use the linearization in equation 9 to transform it into a simple time-varying linear system. The convergence of the algorithms is achieved by studying the stability of this linear system Lessard et al. (2016); Hu et al. (2017). Therefore, the stability condition is naturally used to construct the functional relationship between $\eta_t$ and $L(x_t)$, $\eta_t = \eta(L(x_t))$. This shows that the learning rate is adaptively tuned by $L(x_t)$.

**Predicting the local smoothness.** We secondly predict the local Lipschitz constant $L(x_t)$ by using the current gradient $\nabla f(x_t)$ and the latest gradient $\nabla f(x_{t-1})$. By using the local smoothness in equation 7, $L(x_t)$ on $N_{r_{x_{t-1}}}(x_t)$ is predicted by

$$L(x_t) = \frac{\|\nabla f(x_t) - \nabla f(x_{t-1})\|}{\|x_t - x_{t-1}\| + \epsilon_1}, \tag{10}$$

where $\epsilon_1$ is a parameter to prevent $\|x_t - x_{t-1}\|$ going to zero. This predictive Lipschitz constant $L(x_t)$ is utilized to adjust automatically the learning rate $\eta_t$ for computing the parameter $x_{t+1}$. In the next subsections, we prove that $\eta_t$ is inversely proportional to $L(x_t)$, $\eta_t \propto (1/(L(x_t) + \epsilon_2))$, where $\epsilon_2$ is another parameter to avoid the learning rate to be over large in the later updating process.

**Applying the adaptive learning rate into any stochastic gradient algorithms.** We thirdly use the adaptive learning rate $\eta_t = \eta(L(x_t))$ to update the parameter $x_{t+1}$ in the stochastic gradient algorithm. Overall, Fig. 1 summarizes the proposed adaptive local smoothness method. This method can be applied into the adaptive method based on exponential moving averages (e.g., AdaGrad Duchi et al. (2011), Adadelta Zeiler (2012), RMSProp Tieleman & Hinton (2012), Adam Kingma & Ba (2015), Nadam Dozat (2016) and AMSGrad Reddi & Kumar (2018)), the momentum methods (e.g., HB Polyak (1964), NAG Nesterov (1983) and AccSGD Jain et al. (2017); Kidambi et al. (2018)) and the Quasi-Newton methods (e.g., SQN Byrd et al. (2016)). Next, we will apply PLS into SGD, and AMSGrad Reddi & Kumar (2018) to show its effectiveness. Moreover, due to the limited space, PLS is also used to improve AccSGD Kidambi et al. (2018) in subsection 6.1 in Supplementary Material.

**Remark 1.** *In essence, both PSL and SQN Byrd et al. (2016) have a similarity and a difference. The similarity is that they calculate the Quasi-quadratic differential to accelerate the convergence of stochastic gradient methods. The difference is that SQN employs the Quasi-Newton matrix by*

*equation 4 and PSL uses the predictive local smoothness by equation 10. Compared to SQN, PSL has two advantages. First, PSL requires low computational time and low memory since it only needs to storage the past parameter and gradient, and perform twice subtractions and norm computations. Second, SQN can still be improved by using our PLS, and we empirically verify that PLS-SQN is better than SQN in subsection 4.1.*

**Remark 2.** *Compared to the related local smoothness methods Kpotufe & Garg (2013); Johnson & Zhang (2013); Vainsencher et al. (2015), which require that the loss function is twice differentiable, PLS only requires the differentiable loss function. In addition, the unknown global smoothness is estimated along with the optimization Malherbe & Vayatis (2017), while we provide an effective method to predict unknown local smoothness using equation 10.*

## 3.2 PLS-SGD

In this subsection, we introduce a PLS-SGD algorithm. By using the linearization in equation 9 and computing the expectation, the updating rule of **SGD** is converted into the linear system:

$$x_{t+1} - x^* = ((1 - \eta_t L_{i_t}(x_t)) \otimes I_d)(x_t - x^*), \quad (11)$$

where $i_t$ is sampled in an IID manner from $\Omega = \{1, \cdots, n\}$. Then, the convergence condition of S-GD is obtained by employing the stability condition of the linear system in equation 11, which shows that $x_t$ converges to $x^\star$ at a given linear rate $\rho$. Now we present a linear convergence condition for the SGD as follows.

**Algorithm 1** PLS-SGD.
1: **input:** $\eta_0, \epsilon_1, \epsilon_2$.
2: **initialize:** $x_0$.
3: **for** $t = 1, \cdots, T - 1$ **do**
4:     Randomly pick $i_t$ from $\{1, \cdots, n\}$;
5:     $g_t = \nabla f_{i_t}(x_t)$;
6:     $L_{i_t}(x_t) = \frac{\|g_t - g_{t-1}\|}{\|x_t - x_{t-1}\| + \epsilon_1}$;
7:     $\eta_t = \frac{\eta_0}{(L_{i_t}(x_t) + \epsilon_2)}$;
8:     $x_{t+1} = x_t - \eta_t \nabla f_{i_t}(x_t)$;
9: **end for**

**Theorem 1**[2]. *Consider the linear system in equation 11. Assume that $i_t$ is sampled in an IID manner from a uniform distribution, the assumption 1 holds and there exists an equilibrium $x^* \in \mathbb{R}^d$ such that $\nabla f(x^*) = 0$. Let $\mu = \frac{1}{n} \sum_{i=1}^{n} L_i(x_t)$, and $\nu = \frac{1}{n} \sum_{i=1}^{n} L_i^2(x_t)$. For a fixed linear convergence rate $0 < \rho < 1$, if $\mu^2 > (1 - \rho^2)\nu$ holds, then we have $\frac{\mu - \sqrt{\mu^2 - \nu(1 - \rho^2)}}{\nu} < \eta_t < \frac{\mu + \sqrt{\mu^2 - \nu(1 - \rho^2)}}{\nu}$, and the linear system is exponentially stable, that is, $\|x_t - x^*\|_2 \leq \rho^t \|x_0 - x^*\|_2$.*

Theorem 1 provides a simple condition for the linear convergence of SGD, which benefits from our PLS method. The condition reveals that the functional relationship between $\eta_t$, $\mu$ and $\nu$. In the updating process, since $i_t$ is sampled uniformly at random from $\Omega$, it only uses $\mu = L_{i_t}(x_t)$ and $\nu = L_{i_t}^2(x_t)$. Hence, we have $(1 - \rho)/L_{i_t}(x_t) < \eta_t < (1 + \rho)/L_{i_t}(x_t)$ and set $\eta_t = \eta_0/L_{i_t}(x_t)$, where $\eta_0$ is an initialized learning rate and $1 - \rho \leq \eta_0 \leq 1 + \rho$, and $L_{i_t}(x_t)$ can be predicted by using the equation 10. Similar to $\epsilon_1$, $\epsilon_2$ is another parameter to stop $L_{i_t}(x_t)$ going to zero, and avoid the learning rate to be over large in the latter updating process. In our PLS-SGD algorithm, therefore, the learning rate $\eta_t$ is set to $\eta_0/(L_{i_t}(x_t) + \epsilon_2)$. The adaptive learning rate results in that PLS-SGD has a faster (linear) convergence rate than the traditional SGD. PLS-SGD is summarized in **Algorithm 1**.

## 3.3 PLS-AMSGRAD

Similar to PLS-SGD, we integrate the proposed PLS method into the classical adaptive method based on exponential moving averages, AMSGrad Reddi & Kumar (2018), and propose a PLS-AMSGrad algorithm. The Lipschitz linearization in equation 9 is used to linearize the updating rules in equation 5 of AMSGrad as:

$$\begin{pmatrix} m_{t+1} \\ x_{t+1} - x^* \end{pmatrix} = (A_{i_t} \otimes I_d) \begin{pmatrix} m_t \\ x_t - x^* \end{pmatrix}, \text{ where } A_{i_t} = \begin{pmatrix} \beta_{1t} & (1 - \beta_{1t})L_{i_t}(x_t) \\ -\frac{\eta_t \beta_{1t}}{\sqrt{\widehat{v}_{t+1}}} & 1 - \frac{(1 - \beta_{1t})\eta_t L_{i_t}(x_t)}{\sqrt{\widehat{v}_{t+1}}} \end{pmatrix}, \quad (12)$$

$\widehat{v}_{t+1} = \max\{v_{t+1}, \widehat{v}_t\}$, $v_{t+1} = (\beta_2 \otimes I_d) v_t + ((1 - \beta_2)L_{i_t}^2(x_t) \otimes I_d)(x_t - x^*)^2$. In fact, $m_t$ is the momentum method to manage the velocity of the gradient. Using this linearization, we provide a much simpler convergence analysis of AMSGrad by studying the linear system in equation 12. The linear convergence condition of AMSGrad is described as

---

[2] All the proofs for the Theorems and the linear systems are provided in Supplementary Material.

**Theorem 2.** *Consider the linear system in equation 12. Assume that $i_t$ is sampled in an IID manner from a uniform distribution, the assumption 1 holds and there exists an equilibrium $x^* \in \mathbb{R}^d$ such that $\nabla f(x^*) = 0$. For a fixed linear convergence rate $\rho = \max_t\{\sqrt{\beta_{1t}}\}$, if there exists a $2 \times 2$ positive definite matrix $P \succ 0$ such that*

$$\frac{1}{n}\sum_{i=1}^{n} A_i^T P A_i - \rho^2 P \prec 0, \tag{13}$$

*or the following condition holds, for any $i_t \in \Omega = \{1, \cdots, n\}$,*

$$\frac{\left(1 - \sqrt{\beta_{1t}}\right)\sqrt{\widehat{v}_{t+1}}}{1 + \sqrt{\beta_{1t}}} \frac{1}{L_{i_t}(x_t)} < \eta_t < \frac{\left(1 + \sqrt{\beta_{1t}}\right)\sqrt{\widehat{v}_{t+1}}}{1 - \sqrt{\beta_{1t}}} \frac{1}{L_{i_t}(x_t)}, \tag{14}$$

*then the linear system is exponentially stable, that is, $\left\|\begin{matrix} m_{t+1} \\ x_{t+1} - x^* \end{matrix}\right\|_2 \le \sqrt{cond(P)}\rho^t \left\|\begin{matrix} m_0 \\ x_0 - x^* \end{matrix}\right\|_2$, where $cond(P)$ is the condition number of $P$ and $Cond(P) = \sigma_1(P)/\sigma_p(P)$, where $\sigma_1(P)$ and $\sigma_p(P)$ denote the largest and smallest singular values of the matrix $P$.*

Compared to the convergence analysis of AMSGrad in Reddi & Kumar (2018), Theorem 2 establishes simpler conditions in equation 13 and equation 14 for its linear convergence. The $2 \times 2$ linear matrix inequality (LMI) condition in equation 13 is built by using the control theory (e.g., integral quadratic constraint Lessard et al. (2016); Hu et al. (2017)) to study the stability of the linear system equation 12. It is easily solved by LMI toolbox Boyd et al. (1994). Although the condition in equation 13 is not very clear to the relationship between $\eta_t$ and $L_{i_t}(x_t)$, the condition in equation 13 directly reveals its functional relationship, that is, $\eta_t = \overline{\eta}_t \sqrt{\widehat{v}_{t+1}}/L_{i_t}(x_t)$, where $\frac{1-\sqrt{\beta_{1t}}}{1+\sqrt{\beta_{1t}}} < \overline{\eta}_t < \frac{1+\sqrt{\beta_{1t}}}{1-\sqrt{\beta_{1t}}}$. Based on the equation 5, $\widehat{v}_{t+1}$ tends to zero as it is a linear system, $0 < \beta_2 < 1$, and the gradient goes to zero. For sim-

---

**Algorithm 2** PLS-AMSGrad.

1: **input:** $\eta_0 > 0$, $\{\beta_{1t} > 0\}_{t=0}^{T-1}$, $\beta_2$, $\epsilon_1$, $\epsilon_2$.
2: **initialize:** $x_0 = 0$, $u_0 = v_0 = \widehat{v}_0 = 0$.
3: **for** $t = 1, \cdots, T-1$ **do**
4:     Randomly pick $i_t$ from $\{1, \cdots, n\}$;
5:     $g_t = \nabla f_{i_t}(x_t)$;
6:     $L_{i_t}(x_t) = \frac{\|g_t - g_{t-1}\|}{\|x_t - x_{t-1}\| + \epsilon_1}$;
7:     $\eta_t = \frac{\eta_0}{(L_{i_t}(x_t) + \epsilon_2)}$;
8:     $m_{t+1} = \beta_{1t}m_t + (1 - \beta_{1t})g_t$;
9:     $v_{t+1} = \beta_2 v_t + (1 - \beta_2)g_t^2$;
10:    $\widehat{v}_{t+1} = \max\{v_{t+1}, \widehat{v}_t\}$;
11:    $x_{t+1} = x_t - \eta_t \frac{m_{t+1}}{\sqrt{\widehat{v}_{t+1}}}$;
12: **end for**

---

plification, $\eta_t = \eta_0/L_{i_t}(x_t)$, where $\eta_0$ is an initialized learning rate and $\frac{1-\sqrt{\beta_1}}{1+\sqrt{\beta_1}} < \eta_0 < \frac{1+\sqrt{\beta_1}}{1-\sqrt{\beta_1}}$, since $\beta_{1t} = \beta_1$. Similar to PLS-SGD, $L_{i_t}(x_t)$ is also sampled uniformly at random from $[1, \cdots, n]$, that is, $L_{i_t}(x_t)$ is computed by equation 10, and the learning rate $\eta_t$ is set to $\frac{\eta_0}{(L_{i_t}(x_t)+\epsilon_2)}$ or $\frac{\eta_0}{\sqrt{t}(L_{i_t}(x_t)+\epsilon_2)}$ for avoiding the over-large learning rate in the latter updating process. Thus, PLS-AMSGrad is summarized in **Algorithm 2**.

**Remark 2.** *Based on the Lemma, the $2 \times 2$ LMI in equation 13 is equivalent to the condition in equation 14. The former is obtained by constructing the Lyapunov function in the control theory, while the latter is built by calculating the spectral radius of the weight matrix in the linear system in equation 12, which is defined as the magnitude of the largest eigenvalue of the weight matrix.*

## 4 EXPERIMENTS

In this section, we present empirical results to confirm the effectiveness of the PLS method on linear predictors (convex) and neural networks (nonconvex). We compare PLS-SGD, PLS-SQN, PLS-AMSGrad and PLS-AccSGD with SGD, SQN Byrd et al. (2016), adaQN Keskar & Berahas (2016a), AMSGrad Reddi & Kumar (2018) and AccSGD Kidambi et al. (2018).

### 4.1 PSL FOR CONVEX OPTIMIZATION

In this subsection, we compare PLS-SGD and PLS-SQN with SGD, SQN and adaQN by using convex linear predictor problems (e.g., logistic regression and least squares regression). The code of SGD, SQN and adaQN is downloaded in the github[3] Keskar & Berahas (2016b). Based on this code, we

---

[3]https://github.com/keskarnitish/minSQN

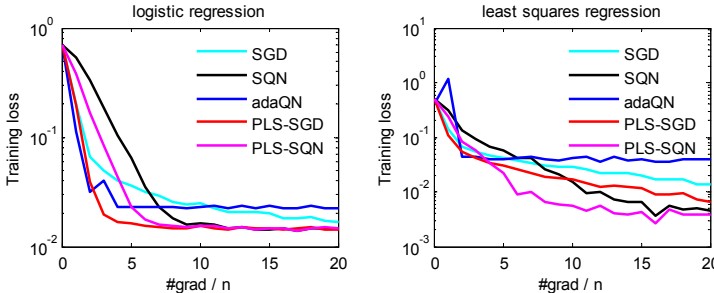

Figure 2: Logistic regression (convex) and least squares regression (convex) on mushroom using SGD, SQN, adaQN, PLS-SGD and PLS-SQN.

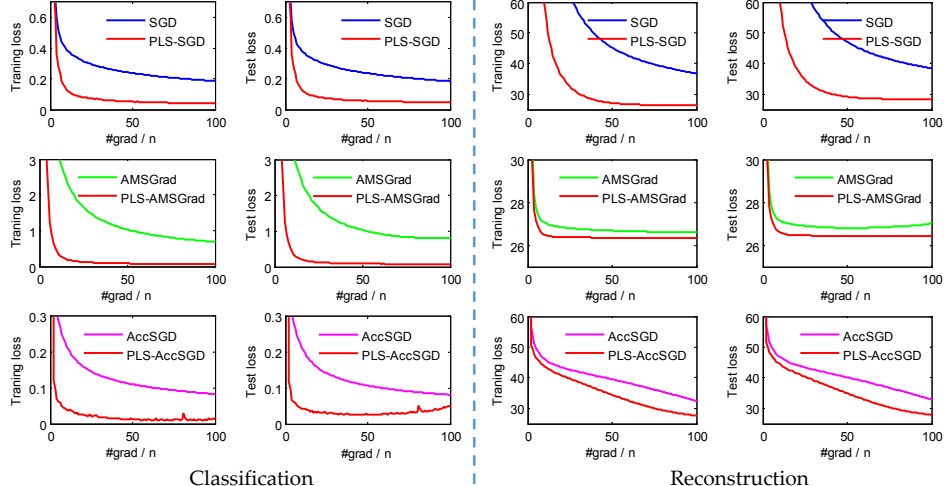

Classification        Reconstruction

Figure 3: Performance comparison of SGD, AMSGrad, AccSGD, PLS-SGD, PLS-AMSGrad and PLS-AccSGD on MNIST using neural networks with two fully-connected hidden layers. The left two columns show the training loss and test loss for classification, while right two columns show the training loss and test loss for reconstruction.

implement PLS-SGD and PLS-SQN. We do experiments on mushroom dataset[4], which contains 8124 data points with 112 dimensions. We use mini-batches of size 256 and tune the learning rate for the best SGD, SQN and adaQN in all experiments. We report the training loss with respect to iterations in Figure 2, and have the following observations. Figure 2 shows the effectiveness of PLS. Specifically, PLS-SGD has faster convergence and lower loss than SGD, and PLS-SQN is also faster convergent than SQN. Moreover, PLS-SGD and PLS-SQN are much better than adaQN.

## 4.2 PSL FOR NONCONVEX OPTIMIZATION

In this subsection, we compare PLS-SGD, PLS-AMSGrad and PLS-AccSGD with SGD, AMSGrad and AccSGD by studying nonconvex multiclass classification and image reconstruction using neural network with least squares regression (LSR) loss and $\ell_2$-regularization. The weight parameters of the neural network are initialized by using the normalized strategy choosing uniformly from $[-\sqrt{6/(n_{in} + n_{out})}, \sqrt{6/(n_{in} + n_{out})}]$, where $n_{in}$ and $n_{out}$ are the numbers of input and output layers of the neural network, respectively. We use mini-batches of size 100 in all experiments.

**Datasets.** MNIST[5] contains 60,000 training samples and 10,000 test samples with 784 dimensional image vector and 10 classes, and CIFAR10[6] includes 50,000 training samples and 10,000 test samples with 1024 dimensional image vector and 10 classes, and 512 dimensional features are extracted by deep residual networks He et al. (2016) for verifying the effectiveness of our methods.

---

[4] https://archive.ics.uci.edu/ml/datasets/mushroom
[5] http://yann.lecun.com/exdb/mnist/
[6] https://www.cs.toronto.edu/~kriz/cifar.html

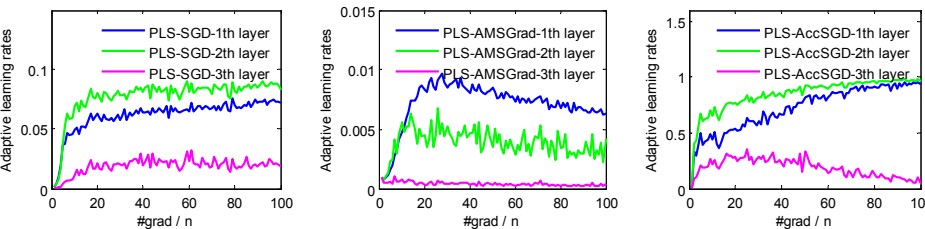

Figure 4: Adaptive learning rates of three layers of neural networks for classification task on MNIST dataset using PLS-SGD, PLS-AMSGrad and PLS-AccSGD.

**Classification.** We train the neural networks with two fully-connected hidden layers of 500 ReLU units and 10 output linear units to investigate the performance of all algorithms on MNIST and CIFA10 datasets. The $\ell_2$-regularization is $1e-4$ (MNIST) and $1e-2$ (CIFAR10). A grid search is used to determine the learning rate that provides the best performance for SGD, AMSGrad and AccSGD. We set the adaptive learning rate $\eta_t = \eta_0/(L_{i_t}(x_t) + \epsilon_2)$ for PLS-SGD, PLS-AMSGrad and PLS-AccSGD, where $\eta_0$ is chosen as $0.001$ or $0.002$. To enable fair comparison, we set typical parameters $\beta_1 = 0.9$ and $\beta_2 = 0.999$ for AMSGrad and PLS-AMSGrad, and set $\kappa = 1000$ and $\xi = 10$ for AccSGD and PLS-AccSGD Reddi & Kumar (2018); Kidambi et al. (2018). For convenience, both parameters $\epsilon_1$ and $\epsilon_2$ are same, $\epsilon = \epsilon_1 = \epsilon_2$, and $\epsilon$ chosen as $0.01$ for PLS-SGD and PLS-AMSGrad, and $0.001$ for PLS-AccSGD.

We report the training loss and test loss with respect to iterations on MNIST in the left two columns of Figure 3. We can see that PLS-SGD, PLS-AMSGrad and PLS-AccSGD preform much better than SGD, AMSGrad and AccSGD. The important reason is that our PLS method can directly adjust the learning rate from a small initialization value to a suitable value. In practice, a large fixed learning rate results in the explosion of the loss of neural network with ReLU using SGD, AMSGrad and AccSGD since the loss will go to infinity when the learning rate is larger than $0.011$ in our experiments. We observe that the learning rate fast increases in the initial stage and slowly varies in the late stage in Figure 4. Moreover, there are similar observations on CIFAR10. Due to the limited space, the losses and learning rate are plotted in Figure 5 in Supplementary Material.

**Reconstruction.** We train a deep fully connected neural network to reconstruct the images in comparison with all algorithms on MNIST dataset. Its structure is represented as $784-1000-500-200-500-1000-784$ with the first and last 784 nodes representing the input and output respectively. In this experiment, we provides the best performance for SGD, AMSGrad and AccSGD by searching from a grid learning rates. The initial learning rate $\eta_0$ is set to $5e$–$7$, $1e$–$2$ and $1e$–$7$ for PLS-SGD, PLS-AMSGrad and PLS-AccSGD, respectively. In addition, we also set to $\eta_t = \eta_0/(\sqrt{t}(L_{i_t}(x_t) + \epsilon_2))$ for PLS-AMSGrad, $\beta_1 = 0.9$, $\beta_2 = 0.999$, $\kappa = 1000$ and $\xi = 10$. Moreover, $\epsilon = \epsilon_1 = \epsilon_2$, and $\epsilon$ chosen as $0.01$ for PLS-SGD and PLS-AccSGD, and $0.1$ for PLS-AMSGrad.

The training loss and test loss with respect to iterations are reported in the right two columns of Figure 3. We can still see that PLS-SGD, PLS-AMSGrad and PLS-AccSGD have significant better performance than SGD, AMSGrad and AccSGD since our PLS method adaptively adjusts the learning rate to prevent the explosion of the LSR loss with large learning rate. The adaptive learning rate is shown in Figure 6 in Supplementary Material. We also observe that the learning rate is initialized a small value, fast increases in the earlier stage and slowly varies in the latter stage.

## 5 CONCLUSIONS

This paper introduced an adaptive local smoothness method for stochastic gradient descent algorithms. This method adjusted automatically learning rate by using the latest gradients to predict the local smoothness. We proposed PLS-SGD, PLS-AMSGrad and PLS-AccSGD algorithms by applying our adaptive local smoothness method into the popular SGD, AMSGrad and AccSGD algorithms. We proved that our proposed algorithms enjoyed the linear convergence rate by studying the stability of their transformed linear systems. Moreover, our proof was significantly simpler than the convergence analyses of SGD, AMSGrad and AccSGD. Experimental results verified that the proposed algorithms provided better performance gain than SGD, AMSGrad and AccSGD.

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

## 6 SUPPLEMENTARY MATERIAL

### 6.1 PLS-ACCSGD

In this subsection, we present a PLS-AccSGD algorithm. Similar to PLS-AMSGrad, our PLS method is integrated into the classical momentum method, AccSGD Jain et al. (2017); Kidambi et al. (2018). Using the Lipschitz linearization in the equation 9, the updating rules in the equation 6 of AccSGD is simply linearized as:

$$\begin{pmatrix} m_{t+1} - x^* \\ x_{t+1} - x^* \end{pmatrix} = (B_{i_t} \otimes I_d) \begin{pmatrix} m_t - x^* \\ x_t - x^* \end{pmatrix}, \tag{15}$$

where $B_t = \begin{pmatrix} \alpha & (1-\alpha)(1-a\eta_t L_{i_t}(x_t)) \\ b\alpha & (1-b)(1-\eta_t L_{i_t}(x_t)) + b(1-\alpha)(1-a\eta_t L_{i_t}(x_t)) \end{pmatrix}$, $\alpha = 1 - \frac{0.7^2\xi}{\kappa}$, $a = \frac{\kappa}{0.7}$, $b = \frac{1-\alpha}{0.7+(1-\alpha)}$, $\xi$ and $\kappa$ are defined in the equation 6. This linearization leads us to provide a much simpler proof for linear convergence analysis of AccSGD by studying the stability of the linear system in the equation 15. We have following Theorem 3 for its convergence condition.

**Theorem 3.** *Consider the linear system in the equation 15. Assume that $i_t$ is sampled in an IID manner from a uniform distribution, the assumption 1 holds and there exists an equilibrium $x^* \in \mathbb{R}^d$ such that $\nabla f(x^*) = 0$. For a fixed linear convergence rate $0 < \rho < \max\left\{1 - \frac{0.7^2\xi}{\kappa}, \max_t\left\{\frac{\kappa(1-\eta_t L_{i_t}(x_t))}{\kappa+0.7\xi}\right\}\right\}$, if there exists a $2 \times 2$ positive definite matrix $P \succ 0$ such that*

$$\frac{1}{n}\sum_{i=1}^n B_i^T P B_i - \rho^2 P \prec 0, \tag{16}$$

*or the following condition holds, for any $i_t \in \Omega = \{1, \cdots, n\}$,*

$$0 < 1 - \frac{0.7^2\xi}{\kappa} < \rho \ \text{ and } \ \left(1 - \rho\frac{\kappa+0.7\xi}{\kappa}\right)\frac{1}{L_{i_t}(x_t)} < \eta_t < \frac{1}{L_{i_t}(x_t)}, \tag{17}$$

*then the linear system is exponentially stable, that is, $\left\|\begin{matrix} m_{t+1} - x^* \\ x_{t+1} - x^* \end{matrix}\right\|_2 \leq \sqrt{cond(P)}\rho^t \left\|\begin{matrix} m_0 - x^* \\ x_0 - x^* \end{matrix}\right\|_2$, where $cond(P)$ is the condition number of $P$.*

Theorem 3 shows that the linear convergence conditions in the equation 16 and the equation 17 are simpler than the convergence analysis of AccSGD Kidambi et al. (2018). Similar to PLS-AMSGrad, the $2 \times 2$ LMI condition equation 16 is built by using the control theory and is easily solved by LMI toolbox Boyd et al. (1994). The condition in the equation 17 directly opens the learning rate $\eta_t$ is a functional relationship with the local smoothness $L_{i_t}(x_t)$, $\eta_t = \frac{\eta_0}{L_{i_t}(x_t)}$, where $\eta_0$ is an initialized learning rate, $1 - \rho\frac{\kappa+0.7\xi}{\kappa} < \eta_0 < 1$. This revels PLS $\eta_0$ can be set to a negative value. The reason is that the eigenvalues of the weight matrix in the system in the equation 15 are $1 - \frac{0.7^2\xi}{\kappa}$ and $\frac{\kappa(1-\eta_t L_{i_t}(x_t))}{\kappa+0.7\xi}$. The stability of the system in the

---

**Algorithm 3** PLS-AccSGD.

1: **input:** $\eta_0, \kappa, \xi \leq \sqrt{\kappa}, \epsilon_1, \epsilon_2$.
2: **initialize:** $x_0, u_0 = 0, \alpha = 1 - \frac{0.7^2\xi}{\kappa}$.
3: **for** $t = 1, \cdots, T-1$ **do**
4:     Randomly pick $i_t$ from $\{1, \cdots, n\}$;
5:     $g_t = \nabla f_{i_t}(x_t)$;
6:     $L_{i_t}(x_t) = \frac{\|g_t - g_{t-1}\|}{\|x_t - x_{t-1}\| + \epsilon_1}$;
7:     $\eta_t = \eta_0/(L_{i_t}(x_t) + \epsilon_2)$;
8:     $m_{t+1} = \alpha m_t + (1-\alpha)\left(x_t - \frac{\kappa\eta_t}{0.7}g_t\right)$;
9:     $x_{t+1} = \frac{0.7}{0.7+(1-\alpha)}\left(x_t - \eta_t g_t\right)$
10:         $+ \frac{1-\alpha}{0.7+(1-\alpha)}m_{t+1}$;
11: **end for**

---

equation 15 needs to satisfy the condition in the equation 17. Similar to PLS-SGD, $L_{i_t}(x_t)$ is also sampled uniformly at random from $[1, \cdots, n]$, that is, $L_{i_t}(x_t)$ is computed by the equation 10. To prevent the over-large learning rate, the learning rate $\eta_t$ is set to $\eta_0/(L_{i_t}(x_t) + \epsilon_2)$ in our PLS-AccSGD, which is outlined in **Algorithm 3**. Following Remark 2, the $2 \times 2$ LMI in the equation 16 is equivalent to the condition in the equation 17.

### 6.2 PROOFS

*Proof of Theorem 1:* First, we prove that SGD in equation 3 is converted into the stochastic linear system in equation 11. By putting the Lipschitz linearization in equation 9 into the equation 3, we

have

$$x_{t+1} = x_t - \eta_t \left(L_{i_t}(x_t) \otimes I_d\right)(x_t - x^*). \tag{18}$$

By adding $-x^*$ into the both sides of the equation 18 and combining like terms, it holds $x_{t+1} - x^* = \left((1 - \eta_t L_{i_t}(x_t)) \otimes I_d\right)(x_t - x^*)$. We thus have the equation 11.

Second, we construct the Lyapunov function $V(x_t) = (x_t - x^*)^T \left(p \otimes I_d\right)(x_t - x^*)$, where $p > 0$, to prove the stability of the system in the equation 11. Defining

$$
\begin{aligned}
\mathbb{E}[\Delta V(x_t)] &= \mathbb{E}[V(x_{t+1}) - \rho^2 V(x_t)] \\
&= \mathbb{E}[(x_{t+1} - x^*)^T \left(p \otimes I_d\right)(x_{t+1} - x^*) - \rho^2(x_t - x^*)^T \left(p \otimes I_d\right)(x_t - x^*)] \\
&= (x_t - x^*)^T \left(\mathbb{E}[(1 - \eta_t L_{i_t}(x_t))^2] - \rho^2\right) \left(p \otimes I_d\right)(x_t - x^*).
\end{aligned} \tag{19}
$$

Then for any $x_t \neq x^*$, $\mathbb{E}[\Delta V(x_t)] < 0$ if $\mathbb{E}[(1 - \eta_t L_{i_t}(x_t))^2] - \rho^2 < 0$, which implies

$$\mathbb{E}[L_{i_t}^2(x_t)]\eta_t^2 - 2\mathbb{E}[L_{i_t}(x_t)]\eta_t + 1 - \rho^2 < 0 \tag{20}$$

Since $i_t$ is sampled in an IID manner from $\{1, \cdots, n\}$, we let $\mu = \mathbb{E}\left[L_{i_t}(x_t)\right] = \frac{1}{n}\sum_{i=1}^{n} L_i(x_t)$, and $\nu = \mathbb{E}\left[L_{i_t}^2(x_t)\right] = \frac{1}{n}\sum_{i=1}^{n} L_i^2(x_t)$. So, the equation 20 is satisfied if we have the condition in Theorem 1 as

$$\mu^2 > (1 - \rho^2)\nu. \tag{21}$$

This implies

$$\frac{\mu - \sqrt{\mu - \nu(1 - \rho^2)}}{\nu} < \eta_t < \frac{\mu + \sqrt{\mu - \nu(1 - \rho^2)}}{\nu}. \tag{22}$$

By using the nonnegativity of the equation 19, we have

$$(x_{l+1} - x^*)^T \left(p \otimes I_d\right)(x_{l+1} - x^*) \leq \rho^2 (x_l - x^*)^T \left(p \otimes I_d\right)(x_l - x^*). \tag{23}$$

Inducting from $l = 1$ to $t$, we see that for all $t$

$$(x_t - x^*)^T \left(p \otimes I_d\right)(x_t - x^*) \leq \rho^{2t}(x_0 - x^*)^T \left(p \otimes I_d\right)(x_0 - x^*), \tag{24}$$

which implies $\|x_t - x^*\|_2 \leq \rho^t \|x_0 - x^*\|_2$, where $p$ is a positive number. The proof is complete. $\square$

*Proof of Theorem 2:* First, we prove that AMSGrad is converted into the stochastic linear system in the equation 12. By putting the Lipschitz linearization in the equation 9 into the equation 5, we have

$$m_{t+1} = (\beta_{1t} \otimes I_d) m_t + (1 - \beta_{1t})(L_{i_t}(x_t) \otimes I_d)(x_t - x^*), \tag{25a}$$

$$v_{t+1} = (\beta_2 \otimes I_d) v_t + (1 - \beta_2)(L_{i_t}(x_t) \otimes I_d)^2 (x_t - x^*)^2. \tag{25b}$$

By adding $-x^*$ into the both sides of $x_{t+1} = x_t - \eta_t \frac{m_{t+1}}{\sqrt{\hat{v}_{t+1}}}$ in the equation 5 and substituting the equation 25a and the equation 25b into the equation 5, it holds

$$
\begin{aligned}
x_{t+1} - x^* &= x_t - x^* - \eta_t \frac{\beta_{1t} m_t + (1 - \beta_{1t})(L_{i_t}(x_t) \otimes I_d)(x_t - x^*)}{\sqrt{\hat{v}_{t+1}}} \\
&= \left(\frac{\eta_t \beta_{1t}}{\sqrt{\hat{v}_{t+1}}} \otimes I_d\right) m_t + \left(\left(1 - \frac{(1 - \beta_{1t})\eta_t L_{i_t}(x_t)}{\sqrt{\hat{v}_{t+1}}}\right) \otimes I_d\right)(x_t - x^*),
\end{aligned} \tag{26}
$$

where $\hat{v}_{t+1} = \max\{v_{t+1}, \hat{v}_t\}$. So, we have the equation 12 by combining the equation 25a with the equation 26.

Second, we construct the Lyapunov function $V(\zeta_t) = \zeta_t^T \left(P \otimes I_d\right) \zeta_t$, where $\zeta_t = \begin{pmatrix} m_t \\ x_t - x^* \end{pmatrix}$ and $P \succ 0$ is a $2 \times 2$ positive matrix, to prove the stability of the system in the equation 12. Defining

$$
\begin{aligned}
\mathbb{E}[\Delta V(\zeta_t)] &= \mathbb{E}[V(\zeta_{t+1}) - \rho^2 V(\zeta_t)] \\
&= \mathbb{E}[\zeta_{t+1}^T \left(A_t \otimes I_d\right) \zeta_{t+1} - \rho^2 \zeta_t^T \left(A_t \otimes I_d\right) \zeta_t] \\
&= \zeta_t^T \left(\left(\mathbb{E}[A_{i_t}^T P A_{i_t}] - \rho^2 P\right) \otimes I_d\right) \zeta_t.
\end{aligned} \tag{27}
$$

Then if the condition in the equation 13 is satisfied, then $\mathbb{E}[\Delta V(x_t)] < 0$ for any $x_t \neq x^*$. By using the nonnegativity of the equation 19, we have

$$\zeta_{l+1}^T (P \otimes I_d) \zeta_{l+1} \leq \rho^2 \zeta_l^T (P \otimes I_d) \zeta_l. \tag{28}$$

Inducting from $l = 1$ to $t$, we see that for all $t$

$$\zeta_t^T (P \otimes I_d) \zeta_t \leq \rho^{2t} \zeta_0^T (P \otimes I_d) \zeta_0, \tag{29}$$

which implies $\left\| \begin{matrix} m_{t+1} \\ x_{t+1} - x^* \end{matrix} \right\|_2 \leq \sqrt{\text{cond}(P)} \rho^t \left\| \begin{matrix} m_0 \\ x_0 - x^* \end{matrix} \right\|_2$, where $\text{cond}(P)$ is the condition number of $P$ and $\text{Cond}(P) = \sigma_1(P)/\sigma_p(P)$, where $\sigma_1(P)$ and $\sigma_p(P)$ denote the largest and smallest singular values of the matrix $P$.

Third, we prove the another condition in the equation 14. If it holds $A_{i_t}^T P A_{i_t} - \rho^2 P \prec 0$ for any $i_t \in \Omega = \{1, \cdots, n\}$, then we have the equation 13. Moreover, $A_{i_t}^T P A_{i_t} - \rho^2 P \prec 0$ is equivalence to a simple condition that the eigenvalues of $A_{i_t}$ is less than $\rho$. Hence, we consider the eigenvalues of $A_{i_t}$ calculated by

$$\lambda_t I - A_t = \begin{pmatrix} \lambda_t - \beta_{1t} & -(1 - \beta_{1t}) L_{i_t}(x_t) \\ \frac{\eta_t \beta_{1t}}{\sqrt{\widehat{v}_{t+1}}} & \lambda_t - \left(1 - \frac{(1-\beta_{1t})\eta_t L_{i_t}(x_t)}{\sqrt{\widehat{v}_{t+1}}}\right) \end{pmatrix} = 0, \tag{30}$$

$$(\lambda_t - \beta_{1t})\left(\lambda_t - \left(1 - \frac{(1-\beta_{1t})\eta_t L_{i_t}(x_t)}{\sqrt{\widehat{v}_{t+1}}}\right)\right) + (1 - \beta_{1t}) L_{i_t}(x_t) \frac{\eta_t \beta_{1t}}{\sqrt{\widehat{v}_{t+1}}} = 0, \tag{31}$$

$$\lambda_t^2 - \left(1 + \beta_{1t} - \frac{(1-\beta_{1t})\eta_t L_{i_t}(x_t)}{\sqrt{\widehat{v}_{t+1}}}\right)\lambda_t + \beta_{1t} = 0, \tag{32}$$

$$\lambda_t = \frac{1 + \beta_{1t} - \frac{(1-\beta_{1t})\eta_t L_{i_t}(x_t)}{\sqrt{\widehat{v}_{t+1}}} \pm \sqrt{\Gamma}}{2}, \tag{33}$$

where $\Gamma = \left(1 + \beta_{1t} - \frac{(1-\beta_{1t})\eta_t L_{i_t}(x_t)}{\sqrt{\widehat{v}_{t+1}}}\right)^2 - 4\beta_{1t}$.

Similar to the Proof of Proposition 1 Lessard et al. (2016), if $\Gamma < 0$, then the magnitudes of the roots satisfy $|\lambda_t| < \sqrt{\beta_{1t}} < \rho = \max_t\{\sqrt{\beta_{1t}}\}$. Then $\Gamma < 0$ implies that

$$\left(1 + \beta_{1t} - \frac{(1-\beta_{1t})\eta_t L_{i_t}(x_t)}{\sqrt{\widehat{v}_{t+1}}}\right)^2 < 4\beta_{1t}, \tag{34}$$

$$-2\sqrt{\beta_{1t}} < 1 + \beta_{1t} - \frac{(1-\beta_{1t})\eta_t L_{i_t}(x_t)}{\sqrt{\widehat{v}_{t+1}}} < 2\sqrt{\beta_{1t}}, \tag{35}$$

$$\frac{\left(1 - \sqrt{\beta_{1t}}\right)^2 \sqrt{\widehat{v}_{t+1}}}{1 - \beta_{1t}} \frac{1}{L_{i_t}(x_t)} < \eta_t < \frac{\left(1 + \sqrt{\beta_{1t}}\right)^2 \sqrt{\widehat{v}_{t+1}}}{1 - \beta_{1t}} \frac{1}{L_{i_t}(x_t)}. \tag{36}$$

The proof is complete. $\square$

*Proof of Theorem 3:* First, we prove that AccSGD is converted into the stochastic linear system in the equation 15. By putting the Lipschitz linearization in the equation 9 into the equation 6, we have

$$m_{t+1} = (\alpha \otimes I_d) m_t + (1 - \alpha)\left(x_t - \frac{\kappa\eta_t}{0.7}\left(L_{i_t}(x_t) \otimes I_d\right)(x_t - x^*)\right), \tag{37a}$$

$$x_{t+1} = \frac{0.7}{0.7 + (1 - \alpha)}\left(x_t - \eta_t\left(L_{i_t}(x_t) \otimes I_d\right)(x_t - x^*)\right) + \left(\frac{1 - \alpha}{0.7 + (1 - \alpha)} \otimes I_d\right) m_{t+1}. \tag{37b}$$

Let $a = \frac{\kappa}{0.7}$ and $b = \frac{1-\alpha}{0.7+(1-\alpha)}$, we have

$$m_{t+1} = (\alpha \otimes I_d) m_t + (1 - \alpha)\left(x_t - a\eta_t\left(L_{i_t}(x_t) \otimes I_d\right)(x_t - x^*)\right), \tag{38a}$$

$$x_{t+1} = (1 - b)\left(x_t - \eta_t\left(L_{i_t}(x_t) \otimes I_d\right)(x_t - x^*)\right) + (b \otimes I_d) m_{t+1}. \tag{38b}$$

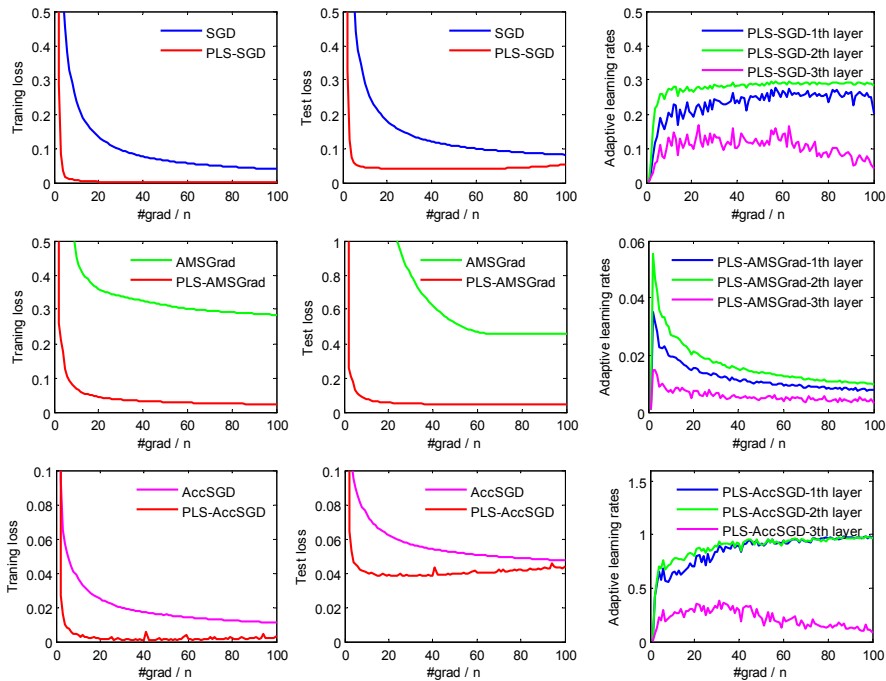

Figure 5: Performance comparison of SGD, AMSGrad, AccSGD, PLS-SGD, PLS-AMSGrad and PLS-AccSGD on CIFAR10 using neural network with two fully-connected hidden layers. The left, middle, and right columns show the training loss, test loss, and adaptive learning rate, respectively.

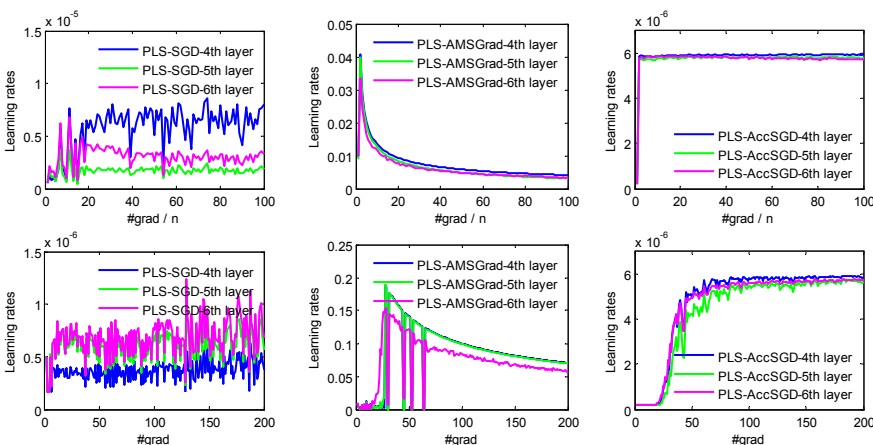

Figure 6: Adaptive learning rates of different layers of neural network for reconstruction using PLS-SGD, PLS-AMSGrad and PLS-AccSGD on MNIST. The down row shows the adaptive learning rate in the first 200 iterations.

By adding $-x^*$ into the both sides of the equation 38a and the equation 38b, and substituting the equation 38a into the equation 38b, it holds

$$m_{t+1} - x^* = (\alpha \otimes I_d)(m_t - x^*) + ((1-\alpha)(1 - a\eta_t L_{i_t}(x_t)) \otimes I_d)(x_t - x^*), \tag{39a}$$

$$x_{t+1} - x^* = ((1-b)(1 - \eta_t L_{i_t}(x_t)) \otimes I_d)(x_t - x^*) + (b \otimes I_d)(m_{t+1} - x^*)$$
$$= (\alpha b \otimes I_d)(m_t - x^*) + (((1-b)(1 - \eta_t L_{i_t}(x_t)) + b(1-\alpha)(1 - a\eta_t L_{i_t}(x_t))) \otimes I_d)(x_t - x^*). \tag{39b}$$

Thus, we have the equation 15 by combining the equation 39a with the equation 39b.

Second, we construct the Lyapunov function $V(\xi_t) = \xi_t^T (P \otimes I_d) \xi_t$, where $\xi_t = \begin{pmatrix} m_t - x^* \\ x_t - x^* \end{pmatrix}, P \succ 0$ is a $2 \times 2$ positive matrix, to prove the stability of the system in the equation 15. Defining

$$
\begin{aligned}
\mathbb{E}[\Delta V(\xi_t)] &= \mathbb{E}[V(\xi_{t+1}) - \rho^2 V(\xi_t)] \\
&= \mathbb{E}[\xi_{t+1}^T (P \otimes I_d) \xi_{t+1} - \rho^2 \xi_t^T (P \otimes I_d) \xi_t] \\
&= \xi_t^T \left( \left( \mathbb{E}[B_{i_t}^T P B_{i_t}] - \rho^2 P \right) \otimes I_d \right) \xi_t.
\end{aligned}
\tag{40}
$$

Then if the equation 16 is satisfied, then $\mathbb{E}[\Delta V(\xi_t)] < 0$ for any $\xi_t \neq 0$. By using the nonnegativity of the equation 40, we have

$$
\xi_{l+1}^T (P \otimes I_d) \xi_{l+1} \leq \rho^2 \xi_l^T (P \otimes I_d) \xi_l.
\tag{41}
$$

Inducting from $l = 1$ to $t$, we see that for all $t$

$$
\xi_t^T (P \otimes I_d) \xi_t \leq \rho^{2t} \xi_0^T (P \otimes I_d) \xi_0,
\tag{42}
$$

which implies $\left\| \begin{matrix} m_{t+1} - x^* \\ x_{t+1} - x^* \end{matrix} \right\|_2 \leq \sqrt{\operatorname{cond}(P)} \rho^t \left\| \begin{matrix} m_0 - x^* \\ x_0 - x^* \end{matrix} \right\|_2$, where $\operatorname{cond}(P)$ is the condition number of $P$.

Third, we certify the another condition in the equation 17. If it holds $B_{i_t}^T P B_{i_t} - \rho^2 P \prec 0$ for any $i_t \in \Omega = \{1, \cdots, n\}$, then we have the equation 16. Moreover, $B_{i_t}^T P B_{i_t} - \rho^2 P \prec 0$ is equivalence to a simple condition that the eigenvalues of $B_{i_t}$ is less than $\rho$. Hence, we consider the eigenvalues of $B_{i_t}$, which is calculated as follows.

By adding a product of $-b$ and the first row of $B_t$ into the second row of $B_t$, $B_t$ is rewritten as $\widehat{B}_t$:

$$
\widehat{B}_t = \begin{pmatrix} \alpha & (1 - \alpha)(1 - a\eta_t L_{i_t}(x_t)) \\ 0 & (1 - b)(1 - \eta_t L_{i_t}(x_t)) \end{pmatrix},
\tag{43}
$$

$$
\lambda_t I - \widehat{B}_t = \begin{pmatrix} \lambda_t - \alpha & -(1 - \alpha)(1 - a\eta_t L_{i_t}(x_t)) \\ 0 & \lambda_t - (1 - b)(1 - \eta_t L_{i_t}(x_t)) \end{pmatrix} = 0.
\tag{44}
$$

The two eigenvalues of $B_t$ is

$$
\lambda_{1t} = \alpha, \quad \lambda_{2t} = (1 - b)(1 - \eta_t L_{i_t}(x_t)).
\tag{45}
$$

Since $\lambda_{1t} > 0$ and $\lambda_{2t} > 0$, we have

$$
0 < \lambda_{1t} = \alpha < \rho, \quad 0 < \lambda_{2t} = (1 - b)(1 - \eta_t L_{i_t}(x_t)) < \rho.
\tag{46}
$$

By substituting $\alpha = 1 - \frac{0.7^2 \xi}{\kappa}$ and $b = \frac{1 - \alpha}{0.7 + (1 - \alpha)}$ into the equation 46, it holds the condition in the equation 17. The proof is complete. $\square$

### 6.3 FIGURES

In the classification experiment, we select the learning rate from $\{0.011, 0.0090.008, 0.007, 0.006, 0.05, 0.004\}$ for providing the best performance of the S-GD, AMSGrad and AccSGD algorithms. In the reconstruction experiment, the learning rate is chosen from $\{6, 5, 4, 3, \}e{-7}$ for SGD and AccSGD and $\{10, 7, 5, 3\}e{-2}$ for AMSGrad due to the explosion of the LSR loss with large learning rate, and select the learning rate from these sets for providing the best performance of the algorithms. To prevent the over-fitting, the learning rate $\eta_t$ is set to $\eta_0 / \sqrt{t}$ for AMSGrad (except MNIST).

Figure 5 reports the adaptive learning rate, the training loss and test loss with respect to iterations on CIFAR10 for classification, while Figure 6 shows the adaptive learning rate with respect to iterations on MNIST for reconstruction.

