# OpenReview forum: "Predictive Local Smoothness for Stochastic Gradient Methods"
_ICLR.cc/2019/Conference_

### Official Review · AnonReviewer3 · 2018-10-19
**Experimental results are too weak**

**Rating:** 4
**Confidence:** 5

**Review:**

In the paper, the authors try to propose an adaptive learning rate method called predictive local smoothness.  They also do some experiments to show the performance.

The following are my concerns:

1. The definition of the L(x_t) is confusing. In (8), the authors define L(x_t), and in (10), the authors give another definition.  Does the L(x_t) in (10) always guarantee that (8) is satisfied?

2. In theorem 1, \mu^2 = \frac{1}{n} \sum_{i=1}^n L_i^2(x_t) + \frac{2}{n^2}  \sum_{i<j}^n L_i(x_t) L_j(x_t) > v. It looks like that \mu > (1-\rho^2) v, no matter the selection of \rho.  Why?

3. How do you compute L_i(x_t)  if x is a multi-layer neural network?

4. The experimental results are too weak. In 2018, you should at least test your algorithm using a deep neural network, e.g. resnet. The results on a two-layer neural network mean nothing.

5. sometimes, you algorithm even diverge. for example, figure 3 second column third row.

---

### Official Review · AnonReviewer2 · 2018-10-30
**Limitedness of contribution and incorrectness of analysis**

**Rating:** 2
**Confidence:** 5

**Review:**

This paper considers the finite-sum optimization problem that is typically seen in machine learning, and proposes methods that adaptively adjust the learning rate by estimating the local Lipschitz constant of the gradient.

The contributions of the paper seem very limited.  The proposed method which estimates the local Lipschitz constant of the gradient, named local predictive local smoothness (PLS) method in the paper (equation (10)), has been proposed in [1] long ago (see equation (11) in [1]) and is very well-known to the community. It is quite surprising that the authors claim to be the first to propose this while completely ignoring previous works.

I also believe that there are major issues with the analysis for the methods. For example, I do not understand how equation (9) could possibly hold for general functions, and how it could be possible to transform their method into the linear system in (11). Therefore I do not think this paper is technically correct.

In summary, I believe this paper is limited in its contribution and also has major issues in terms of technical correctness, and is well below the standard for ICLR.

Reference:

[1] Magoulas, G. D., Vrahatis, M. N., & Androulakis, G. S. (1997). Effective backpropagation training with variable stepsize. Neural networks, 10(1), 69-82.

---

### Official Review · AnonReviewer1 · 2018-11-10
**The Analysis seems incorrect**

**Rating:** 3
**Confidence:** 3

**Review:**

The paper proposes to use an estimate of the 'local' smoothness constructed by taking the difference of the gradients along the previous step. This is a simple idea and has been considered before in literature. The authors seem to take a very simplistic approach to the problem which seems to not work at all in high dimensions. I am reasonable certain that the analysis is incorrect as it is impossible to get linear convergence via SGD or even with GD in general settings. Looking at the proof which is written in a very unreadable way reveals that they make multiple assumptions which holds basically in the case of a quadratic and then further only in one dimension. In which case such a rate with GD is trivial.

So the theory is blatantly wrong. Regarding the experiments they also look shaky at best and sometimes they diverge. I believe the paper is much below standard for ICLR.

---

### Official Review · AnonReviewer4 · 2018-11-12
**Unrealistic assumptions, trivial theory**

**Rating:** 2
**Confidence:** 4

**Review:**


# Unrealistic assumptions and trivial theory

This papers proposes a method to adjust the learning rate of stochastic gradient methods. The problem is of great importance but the theoretical results and presentation contain many issues that make the paper unfit for publication.

The main issue that I see is that the assumption made are unrealistic and make the theory trivial. First, for gradient descent, the authors assume that the gradient is of the form L(x_t) (x_t - x*). Under this assumption, gradient descent converges on a single step with step size 1 / L(x_t). In the stochastic setting, they assume that *each* stochastic gradient is of the form L_i(x_t) (x_t - x*), Eq. (11). Again, SGD in this scenario converges in a single iteration with step size 1 / L_i(x_t).

No wonder in this scenario the authors are able to obtain linear convergence of SGD to arbitrary precision (which is known to be impossible even for quadratics).


# Other Issues

* Motivation of Eq. (9) is not discussed in sufficient detail. It is unclear to me how to obtain (9) from (7) as the authors mention. Regarding notation, L(x_t) is a scalar, hence (9) could be written more simply as \nabla f(x_t) = L(x_t) (x_t - x*). Why the need for the Kronecker product?

* The authors should clearly state what are the assumptions in the theorem statement. For theorem 1 these are not clearly stated, and phrases like "Theorem 1 provides a simple condition for the linear convergence of SGD" give the wrong impression that the Theorem is widely applicable.


# Minor
  * Belo Eq. (10): "where \epsilon_1 is a parameter to prevent ||x_t - x_{t-1}|| going to zero: . I guess what the authors meant is to prevent *the denominator* going to zero, you do want ||x_t - x_{t-1}|| to go to zero as you approach a stationary point

---

### Meta-Review · Area_Chair1 · 2018-12-13
**Issues with the analysis**

**Confidence:** 5
**Recommendation:** Reject

**Metareview:**

Dear authors,

All reviewers pointed to severe issues with the analysis, making the paper unsuitable for publication to ICLR. Please take their comments into account should you decide to resubmit this work.